# Unexpected Benefits of Multiport Synchrotron Microbeam Radiation Therapy for Brain Tumors

**DOI:** 10.3390/cancers13050936

**Published:** 2021-02-24

**Authors:** Laura Eling, Audrey Bouchet, Alexandre Ocadiz, Jean-François Adam, Sarvenaz Kershmiri, Hélène Elleaume, Michael Krisch, Camille Verry, Jean A. Laissue, Jacques Balosso, Raphaël Serduc

**Affiliations:** 1INSERM UA7, STROBE, 71 Av. des Martyrs, 38000 Grenoble, France; eling@esrf.fr (L.E.); Audrey.bouchet@inserm.fr (A.B.); Ocadiz@esrf.fr (A.O.); adam@esrf.fr (J.-F.A.); keshmiri@esrf.fr (S.K.); Helene.elleaume@inserm.fr (H.E.); 2European Synchrotron Radiation Facility, 71, Av. des Martyrs, 38000 Grenoble, France; krisch@esrf.fr; 3Radiotherapy Department, Grenoble University Hospital, 38000 Grenoble, France; CVerry@chu-grenoble.fr (C.V.); j.balosso@baclesse.unicancer.fr (J.B.); 4Institute of Anatomy, University of Bern, 3000 Bern, Switzerland; jean-albert.laissue@pathology.unibe.ch; 5CLCC Francois Baclesse, 14000 Caen, France

**Keywords:** synchrotron microbeam radiation therapy, brain tumor control, dose equivalence, normal tissue sparing

## Abstract

**Simple Summary:**

We unveiled the potential of an innovative irradiation technique that ablates brain cancer while sparing normal tissues. Spatially fractionating the incident beam into arrays of micrometer-wide beamlets of X-rays (MRT for Microbeam Radiation Therapy) has led to significantly increased survival and tumor control in preclinical studies. Multiport MRT versus conventional irradiations, for the same background continuous dose, resulted in unexpectedly high equivalent biological effects in rats that have not been achieved with any other radiotherapeutic method. These hallmarks of multiport MRT, i.e., minimal impact on normal tissues and exceptional tumor control, may promote this method towards clinical applications, possibly increasing survival and improving long-term outcomes in neuro-oncology patients.

**Abstract:**

Delivery of high-radiation doses to brain tumors via multiple arrays of synchrotron X-ray microbeams permits huge therapeutic advantages. Brain tumor (9LGS)-bearing and normal rats were irradiated using a conventional, homogeneous Broad Beam (BB), or Microbeam Radiation Therapy (MRT), then studied by behavioral tests, MRI, and histopathology. A valley dose of 10 Gy deposited between microbeams, delivered by a single port, improved tumor control and median survival time of tumor-bearing rats better than a BB isodose. An increased number of ports and an accumulated valley dose maintained at 10 Gy delayed tumor growth and improved survival. Histopathologically, cell death, vascular damage, and inflammatory response increased in tumors. At identical valley isodose, each additional MRT port extended survival, resulting in an exponential correlation between port numbers and animal lifespan (r^2^ = 0.9928). A 10 Gy valley dose, in MRT mode, delivered through 5 ports, achieved the same survival as a 25 Gy BB irradiation because of tumor dose hot spots created by intersecting microbeams. Conversely, normal tissue damage remained minimal in all the single converging extratumoral arrays. Multiport MRT reached exceptional ~2.5-fold biological equivalent tumor doses. The unique normal tissue sparing and therapeutic index are eminent prerequisites for clinical translation.

## 1. Introduction

Glioblastoma multiforme (GBM) is the most common type of human primary brain malignancies (48.3% [1]) and has the poorest prognosis. The multimodal approach of surgical resection, radiotherapy (RT), and chemotherapy with temozolomide [2], known as the Stupp regimen, leads to a median survival time (MST) of only 14.6 months and has not significantly improved over the last 10 years [3]. Aggressive adjuvant treatment strategies also elicit severe side effects on normal tissues. Conventional radiotherapy (RT) often leads to complications such as neurocognitive toxicity and leukoencephalopathy, especially when administered as whole-brain RT [4]. Temporal fractionation of a 60 Gy total dose, delivered in daily sessions of 2 Gy over 6 weeks, remains the standard of care [5]; however, therapeutic efficacy is limited, progression-free survival does not exceed 7 months, and fewer than 7% of patients survive longer than five years [1]. Unfortunately, most radiotherapeutic protocols for GBM management have not significantly evolved for years nor have they improved post-treatment quality of life. 

Microbeam Radiation Therapy (MRT), an innovative, radically new approach in radiation oncology, has been developed at synchrotron X-ray sources, and mainly for the last 2 decades, at the European Synchrotron Radiation Facility (ESRF) in Grenoble, France. The physical characteristics (very high dose rate, low energy, quasi-null divergence) of synchrotron-generated X-rays enable the spatial fractionation of incident beams into multiple micrometer-scaled microbeams spaced at 200 to 400 micrometers. The dose deposited in the path of these microbeams (peak dose) can be as high as hundreds of gray, while the dose diffused in-between the microbeams (valley dose) amounts to only 1–5% of the peak dose. Normal brain tissue is eminently tolerant to MRT [6]; cell loss is confined to microbeam paths without disruption of mature vasculature, maintaining the continuous perfusion of normal tissues [7,8], even a long time after exposure [9,10]. In contrast, preferential damage to immature tumor vessels [11] reduces oxygen and nutrient supply, and causes tumor necrosis [12,13]. MRT has been shown to significantly improve tumor control in preclinical experiments compared to conventional RT at MRT valley doses similar to those of conventional (Conv.) homogeneous “Broad Beam” (BB) irradiations [11,14].

To date, all preclinical MRT experiments on tumor models have investigated the effects of 1 single or 2 crossing orthogonal irradiation ports (or beam trajectories). The influence of adjustable irradiation parameters (spectrum, microbeam width, spacing, etc.) has been studied and differential responses between tumor control and normal tissue sparing have been found [13,15,16,17]. However, one critical parameter, namely the number of ports, has never been systematically investigated until now. Data derived from equivalent MRT valley doses delivered by either one or two crossing ports have shown an accumulation of cellular/vascular microbeam-induced lesions within the tumors, which might account for the increased MST of animals [18]. Based on these results, we hypothesized that an increasing number of incident MRT arrays, while keeping the same cumulated valley dose, might significantly improve tumor control due to increased numbers of high-dose microbeams and spike-like dose hot spots in the target volume, while minimizing normal tissue dose in the single path of each array. We predicted a non-linear relationship between tumor control and the number of MRT beam trajectories, and thus an improvement in the therapeutic index.

In this study, we assessed the effects of MRT delivered to 9LGS tumor-bearing or tumor-free rats through up to five irradiation ports (MRT2 to MRT5); we compared these results with responses following crossed BB (BB2) exposures. Our data confirm that an increasing number of MRT ports provide a non-linear improvement in tumor control with unexpectedly high anti-tumor equi-effective dose (EquiED), while sparing normal brain tissues. These findings highlight the potential promise of this new irradiation modality for clinical applications.

## 2. Materials and Methods

Procedures related to animal care comply with the Guidelines of the French Government (licenses #380325/#390321, authorized labs A3818510002/A3851610008/A3851610004).

### 2.1. Behavioral Tests of Normal Rats

Normal rats (*n* = 22) were tested for cognitive and motor function 0.5, 2, 6, and 12 months post irradiation (p.i.). Results were compared to behavioral patterns of unirradiated normal control rats (*n* = 10). Testing included an open-field (OF) test; novel object recognition (NOR) tasks; a motor function and coordination test (Rotarod). Procedures were carried out according to previous protocols [19]. Detailed procedures are described in Appendix A.

### 2.2. Tumor Cell Implantation and Randomization

The 9L gliosarcoma cells (*n* = 10^4^, Sigma-Aldrich, St. Quentin Fallavier, France) were implanted in the right caudate nucleus of ten-weeks-old male Fisher rats (*n* = 160, Charles River Laboratories, Ecully, France) as previously described [20]. Nine days after implantation, all rats underwent T_2_-weighted magnetic resonance imaging (MRI). The animals with comparable tumor volume and locations were randomized into groups (*n* = 6). Group sizes for all experimental conditions are summarized in Figure 3C and Appendix A.

### 2.3. Radiation Sources, Dosimetry, and Treatments

MRT was performed 10 days after tumor inoculation at ID17 at the European Synchrotron Radiation Facility (ESRF) in Grenoble (France). Details of the irradiation setup and beam properties are given in [15]. Briefly, rats were exposed to single (MRT1) or to multidirectional MRT using 2 to 5 irradiation ports, the latter applied in microbeam mode (8 × 8 mm^2^ irradiation field, 19 microbeams, width 50 µm, spacing 400 µm, median beam energy of 90 keV) or, with 2 ports only, in Broad Beam mode (BB2, 8 × 8 mm^2^ irradiation field). Both irradiation modalities were performed at a very high dose rate (12–16 kGy.s ^−1^). Radiation doses were calculated using the hybrid algorithm developed by Donzelli et al. [21]. The peak and valley dose maps were extracted according to the method detailed in Ocadiz et al. [22]. The dose-volume histograms (DVHs) were calculated on these maps. We arbitrarily fixed the upper limit of cumulated valley dose at 10 Gy for the MRT configurations, corresponding to the 10 Gy BB2 dose delivered to the whole tumor; this led to a mean valley dose in the tumor of 9.35 Gy (8.5–10 Gy). Entrance dose prescriptions for the different groups of rats are summarized in Figures 1C and 3C, and Appendix A. The number of days (*n*) elapsed were noted in the text as follows: Tn (p.i.).

A 5-beam MRT treatment plan has been calculated on a patient bearing a 1 cm diameter brain metastasis from a primary lung cancer. The Gross Tumor Volume (GTV) as well as the Planning Target Volume (PTV) and whole brain (Organ at Risk, OAR) have been contoured. The dose calculations were performed on CT scan images using the hybrid algorithm [21]. The peak and valley dose maps and peak and valley dose-volume histograms were then extracted for the GTV, PTV, and OAR.

### 2.4. Animal Monitoring after Irradiation

Anatomic MRI at 7, 14, and 21 days were performed after irradiation on tumor-bearing 9L rats in order to follow the evolution of tumor growth. T_2_ Turbo RARE images were acquired in axial and horizontal planes. MRI of normal rats (no tumor; MRT2, MRT5, BB2) was performed on the same magnet at 2, 6, and 12 months p.i. using T_2_-weighted axial images. A brain diffusion map and 3D T_2_ star map MGE were also acquired 12 months p.i. in normal animals.

### 2.5. Pathology and Immunohistology of Brain Sections

Tumor-bearing animals not included in survival studies were sacrificed and their brains were removed 7 or 14 days p.i. The brains of normal, not tumor-bearing rats were removed one year p.i. Two coronal cryosections (18 µm thick) of 4 animals in each group and at each sample time point were immunolabeled as described in [13] and detailed in the Appendix A.

## 3. Results

Irradiation geometries and dose–volume histograms (DVHs) are shown in Figure 1A,B. The whole brain DVHs for valley doses (Figure 1B) are higher for MRT, compared with broad beam doses, because of the additional scatter dose produced by the peaks.

### 3.1. Effects of Multiport MRT on Tumor Free Animals

#### 3.1.1. Neurological Changes and Survival after Microbeam Irradiations

T_2_-weighed MR images acquired 2, 6, and 12 months p.i. in normal rats (details in Figure 1C) showed zones of altered signal in the tissue volume covered by the MRT field, while BB2 irradiation did not induce tissue damage detectable by MRI (Figure 1D). Apparent diffusion coefficients (ADC) revealed significantly decreased whole brain water diffusion after BB2 irradiation compared with MRT2/5, or in untreated brains (*p* < 0.05, not shown). Whole brain T_2_* fits were significantly lower after MRT compared with BB2 therapy or without treatment (*p* < 0.005, Figure 1E). Significantly different T_2_* values were seen between target (RCN) and contralateral left caudate nucleus (LCN) in MRT irradiated animals (*p* < 0.001, Figure 1E). T_2_* relaxation time in the RCN of MRT2/5 rats was significantly shorter than in BB2 and control rats (*p* < 0.001), whereas ADC values in the target did not differ between groups, nor from those of the contralateral LCN (cf. Figure 1D). One hundred percent survival was reached after BB2 and MRT2, whereas two out of eight animals died 11 and 12 months after MRT5 (75% survival rate; *p* = 0.1, Figure 1F).

#### 3.1.2. Multiport MRT Modified Normal Rat Ambulation

Results of the novel environment exploration (open-field test) are shown in Figure 1G. A change in general ambulation, observed after microbeam exposures, was not significant. MRT5-treated rats covered a longer distance in the center zone two weeks p.i. (*p* = 0.05), but not at the next test point. No significant differences in novel object recognition were observed between irradiated and non-irradiated animals (Figure 1I). Motor function and coordination were not significantly altered after multiport MRT versus controls (Figure 1J). However, animals irradiated through five MRT ports had significantly higher running scores between 2 and 4 weeks p.i. Analytic details of behavioral changes are given in the Appendix A.

#### 3.1.3. Histopathology Revealed Sparing of Normal Tissues Irradiated by a Single Array

Results of the histologic analysis of brain sections sampled 1 year p.i. are shown in Figure 2. The radio-induced changes in normal brain tissue were located outside the intersecting regions, where only one array of microbeams was deposited. After BB2 irradiation, no changes in tissue structure were seen on HE-stained sections (Figure 2A) nor on immunofluorescent-labeled images (Figure 2B–E). After MRT, microbeam stripes could be distinguished in MRT2 brains in otherwise intact tissue structures; no microcalcifications were observed in both MRT groups (Figure 2A). All applied immunomarkers reacted comparably to those in the non-irradiated control group. They indicated preserved microvasculature (Figure 2B), moderate influx of macrophages (Figure 2C), and unchanged oligodendrocyte (Figure 2D) and neuronal density (Figure 2E). Total cell densities did not change after any of the irradiation configurations used (Figure 2F) and neither BB2 nor MRT2/5 modified the numbers of blood vessels (Figure 2G). A limited increase in macrophage density was depicted with an increasing number of MRT ports. The numbers of neurons (Ctrl vs. MRT5 *p* = 0.51, Figure 1H) remained similar in all groups. Histopathologic changes of normal tissue in the target region are displayed in Appendix A.

### 3.2. Effects of Multiport MRT on Brain Tumors

#### 3.2.1. At Equal Valley Dose, Additional MRT Ports Non-Linearly Improved 9L Tumor Control

The effects of additional MRT ports—from 1 to 5—on the 9L tumor control were evaluated after a 10 Gy cumulated valley dose (Figure 3A). To aptly compare the dose effect of MRT and BB2 modes, we limited the cumulated valley dose to 10 Gy for any of the MRT configurations, i.e., to a mean valley dose for 9.35 Gy [8.5–10 Gy] in the MRT mode. For detailed doses and DVHs, see Figure 3B,C. Tumor volumes on days 14 and 21 p.i. demonstrated that the MRT mode significantly improved tumor control in all geometries used. Even after only one array of microbeams, brain tumors were two times smaller after MRT than after BB2 at T14 (*p* = 0.0033, Figure 3D). Two weeks after MRT from five ports, tumors were 10.6 times smaller (*p* < 0.0001). Tumor control increased exponentially with every additional MRT port (Figure 3D). Figure 3E,F report dose equivalences in terms of tumor control between BB2 and MRT: a 10 Gy valley dose MRT delivered via one or five ports was equivalent to 16.4 ± 2.2 or 27.3 ± 0.5 Gy BB2 exposures on T14, and to 13.1 ± 2.2 or 22.3 ± 1.0 Gy BB2 on T21, respectively.

#### 3.2.2. MRT Increased Median Survival of Tumor-Bearing Rats

MST of 9L bearing rats are shown in Figure 3G as typical sigmoidal tumor response to increasing radiation doses, with a plateau at 35.06 ± 2.03 Gy. A BB2 irradiation of 10 Gy with two cross-fired beams significantly improved MST compared with controls, from 10.5 to 18 days p.i., respectively (*p* < 0.002). Lifespans were extended with increasing numbers of MRT ports. Figure 3G (center) illustrates the exponential increase in MST with the increasing number of MRT ports, delivering a 10 Gy cumulated valley dose. When MST is plotted in function of number of ports (Figure 3G, right), the MRT fit curve would reach a plateau at 8.22 ± 0.39 MRT ports, i.e., the theoretical number of ports that would improve MST corresponding to a 35 Gy BB2 irradiation.

#### 3.2.3. Multiport MRT Induced Pronounced Histopathologic Changes in 9L Tumors

##### Effects of Crossed BB2 Irradiation on 9L Gliosarcoma

As depicted on HE sections (Figure 4A–F), tumors displayed typical 9L gliosarcoma cell features [23]. BB2 irradiation slowed tumor growth compared with controls (Figure 4G), correlating with increased numbers of γH2AX-positive cells (Figure 4B,H) and reduced Ki67-positive cells numbers (Figure 4C,I). Tumor blood volume fraction (BV, Figure 4D,J) was not modified by BB2 exposure; the latter led to a progressive invasion of macrophages between T7 and T14 (*p* < 0.02, Figure 4E,K) while microglial cells did not significantly infiltrate 9L tumors (Figure 4F,L).

##### MRT2/5 Effects on 9L Gliosarcoma

MRT significantly reduced tumor surface areas on histological sections (Figure 4A,G) compared with BB2 irradiation. MRT-irradiated 9L tumors did not grow between T7 and T14 (Figure 4G) while BB2 tumors recurred (T7 vs. T14, *p* < 0.0001). Figure 4B,C,H,I show that MRT induced significantly more DNA damage 1 week p.i. (γH2AX+ cells, at T7 *p* < 0.0001 vs. Ctrl and *p* < 0.05 vs. BB2) and significantly reduced tumor proliferative activity (Ki67+ cells, at T7 *p* < 0.002 vs. Ctrl) for 2 weeks after irradiation. Vascular effects were only detected after MRT5: a decrease in tumor BVf versus controls could be observed at T7 (Figure 4D,J). Immune cells infiltration occurred not only in marginal but also in central areas of tumors in both MRT groups (Figure 4E,F,K,L). For instance, the surface fraction invaded by macrophages (CD68) significantly increased at T7 after MRT5, compared with control and BB2-irradiated tumors (*p* < 0.0005); this increased even further until T14 after MRT5 (T7 vs. T14 *p* < 0.02, Figure 4K). Microglial invasion (CD11b fraction) was significantly higher at both time points after MRT, versus that seen in BB2-treated tumors (*p* < 0.005, Figure 4L).

### 3.3. Simulation of an MRT Treatment of Brain Metastasis in a Human Patient

Figure 5 shows the peak and valley dose maps for a patient bearing a 1 cm diameter brain metastasis located at 5–8 cm depth (Figure 5A) treated with a 15–16 mm diameter PTV. PTV, GTV, and whole brain DVHs (Figure 5B) show the correct coverage of the GTV with cumulated valley doses ranging between 9 and 10 Gy, whereas the peaks (50 to 80 Gy at target) will cumulatively generate 250–300 Gy hot spots in the tumor. For each port, peak entrance doses were between 150 and 180 Gy and valley doses did not exceed 6 Gy.

## 4. Discussion

MRT controls malignant tumors more efficiently than homogeneous broad X-ray beams and is well tolerated by normal tissues. The present study demonstrates that increasing the number of incident ports not only reduces the dose deposited in normal tissues, but significantly enhances tumor control with dose equivalence factors for Conv. BB ranging from 1.4 to 2.5 and exponentially improves survival times of rats bearing tumors. 

Normal brain tissues’ tolerance of MRT has been studied using behavioral tests and histopathology. No significant changes were observed in tissues irradiated with a single-port trajectory; 5-port irradiation elicited toxicity only in the target region but never in unidirectionally irradiated brain regions. Damage was evident on MR images by signs of hemorrhages detected as soon as 2 months p.i. and, to a lower extent, in the MRT2 group, starting 6 months p.i. Remarkably, vascular networks had recovered from radiation-induced damages at 1 y p.i., when ADC values in the target no longer differed between MRT-irradiated and control animals. Irradiation did not modify motor coordination nor memory abilities; notably, 10 Gy MRT through 5 ports reduced anxiety-like behavior, shown by increased locomotor activity and explorative behavior in the open-field test (Figure 1G). Pathology confirmed that the extent of tissue damage increased inside the target (Appendix A) with the number of MRT ports, but not outside that target (Figure 2), where cell loss was exclusively detected in the path of microbeams. These results are in line with the ones described by Laissue et al. [18] and in Bouchet et al. [12,13], which all showed histological damage scores and microcalcifications, a common sign of radiation-induced damage [24,25], concentrated in crossfired regions, whereas unidirectionally irradiated normal brain tissues displayed low damage scores. Please take note that dosimetric characteristics, for normal rats and for tumor irradiations, particularly peak to valley dose ratios, were strictly reproduced in the present work. Because conformal irradiations are not yet technically feasible in small animals, an oversized, 8 × 8 × 8 mm^3^, irradiation field was used, which would cover about 100 tumor volumes (≈5.2 mm^3^ at T_irr_). Despite this overexposure by an MRT-equivalent irradiation, the survival of normal rats without tumors remained quite similar to that of non-irradiated rats. The dose–volume histogram (DVH, Figure 1B) shows that, for the 5 MRT ports configuration, nearly the whole rat brain received 5 Gy while BB2 irradiation delivered 5 Gy to only 50% of the brain, and completely spared 20% of normal tissues. Crossfired MRT arrays, delivered by oversized radiation fields, are certainly more neurotoxic than a BB2 irradiation at equivalent valley doses. However, conformal irradiation fields, closely adapted to the tumor, are needed to drastically reduce MRT toxicity. Another phenomenon requires further investigations: how narrowly must the “star effect” in the normal brain, created around the target by the intersected arrays outside the targeted zone, be confined to the tumor margins? In other words, what is the maximum volume of normal brain around the tumor that will tolerate such radiation toxicity?

More than one decade ago, MRT has demonstrated to be more efficient for tumor control than Conv. BB irradiations [14,26], a fact clearly confirmed in the present study: when using a valley dose equivalent to a Conv. BB dose (10 Gy), unidirectional MRT halves the tumor volume despite the fact that 7/8 of the tumor cells received the BB-like dose, but only 1/8 a lethal dose by MRT. This shows that MRT antitumor effects are not directly due to intrinsic physical properties of the X-ray beam, but that associated biological processes underlie tumor response to high dose microbeams. Vascular effects have been described [12,13] and differential molecular pathways have been identified [27], but many of the radiobiological factors governing those specific effects of MRT are still unknown. An important role of immune response after MRT exposures has been suggested [14,27,28,29]; previous research showed that a combination of MRT with gene-mediated immunoprophylaxis significantly increases the survival of tumor-bearing rats [29]. The massive infiltration of macrophages in the tumor, as seen in our study, deserves further investigations as it might also participate in tumor control after MRT. At identical valley dose, each supplementary MRT port extended MST of 9L tumor rats by 2 to 3 days. Furthermore, a strong non-linear correlation between the number of MRT ports and MST has been found (Figure 3F–H). These results highlight, for the first time, the exponential increase in survival time by multidirectional MRT, while the 10 Gy valley dose, a Conv. BB dose equivalent, remains untouched. In terms of survival, MRT2 was as efficient as an approximately 17 Gy BB2 dose. MRT5 led to MST comparable to that achieved by a 24 Gy BB2 fraction. By extrapolation, 8 MRT ports, with a 10 Gy cumulated valley dose, would be as efficient as a BB2 35 Gy exposure and ablate about 80% of 9L tumors. The EquiED of MRT thus reached unexpected values (up to ×2.4 the valley dose, Figure 3H), an effect that has not been achieved by any other radiotherapy method based on pure ballistic effects.

Multidirectional MRT with extremely high EquiED will be limited by radiation-induced neurotoxicity. Presently, up to 5 ports did probably contribute to an aggressive destruction of tumor cells and a release of toxic degradation products able to induce a tumor lysis syndrome. Immune system activation and edema may initiate and promote inflammatory processes [30]. Cortical neurons may react to such stimuli by uncontrolled discharges, which probably led to seizures in some of the tumor-bearing rats, but importantly, never did that in normal animals. Early toxicity can be reduced by: (i) conforming irradiation beams to tumor size and shape; (ii) using steroids to control brain edema, as is routinely done in clinical practice; (iii) debulking the tumor surgically before irradiation; and (iv) temporally fractionating the MRT dose/port delivery. Temporal fractionation of multiport MRT, not investigated in vivo, may drastically reduce normal brain tissue damage and toxicity related to tumor cell necrosis.

The current radiotherapy dosage for GBM has been set to 60 Gy in 30 fractions, and the design of the Stupp trial has not been modified for years [31]. Human GBM mostly relapses locally, thus an MRT-boost delivered through multiple ports might significantly improve tumor control while decreasing out-of-target neurotoxicity [22,32,33]. Some studies mention the relevance of a radiation boost delivered to hyperactive tumor areas (as detected by PET) or hyperintensity regions on T_2_-weighted MR images [3]. MRT could be used at first in patients as a boost or a limited part of a hypofractionated treatment [33], in which doses larger than 2 Gy per fraction are commonly used (e.g., brain metastasis, 3 × 11 Gy; GBM boost regimen, 46 + 14 Gy or 50 + 10 Gy [3]). Experimental data suggest that such boost doses made by MRT could be significantly more efficient than conventional treatments [33] and particularly relevant for such lesions, since tumor control improves with an increasing number of ports, while cumulated valley doses remain rather low (note that only 1.0 Gy (MRT2) or 0.34 Gy (MRT5) per valley dose per port would be sufficient for the control of 9LGS as a 4 Gy BB2 exposure, see Appendix A). The so-called “cone-down” practice [3] or even more the principle of dose painting may be relevant for MRT that can deliver high EquiED of radiation to “high risk” tumor regions.

Despite the use of low energy synchrotron-generated photons (~100 keV), our simulations performed on a clinical case (target at 5–8 cm depth) suggest that MRT clinical transfer is feasible and realistic in a medium-term time scale. In the proposed MRT treatment on patients, the average energy will be increased to 120 keV, in order to improve the PVDR in depth [34]. Peak entrance doses required to deliver a 10 Gy cumulated valley dose reach a maximum of 180 Gy, whilst entrance valley doses do not exceed 6 Gy. These entrance radiation doses do not exceed the tissue tolerances for normal brain and skin animal models and in humans for small fields like the ones used in this study. Such irradiations could, in a clinical context, be as effective as a 25 Gy Conv. BB fraction.

## 5. Conclusions

To conclude, we emphasize that multiport MRT reached unexpectedly high equi-effective doses (~2.5 fold), compared with conventional BB irradiation. This can be attributed to the increase in microbeams and/or in the spike-like high dose spots in the target volume with an increasing number of MRT ports. According to the results of the current study, the balance between tumor control probability and normal tissue complication probability (TCP/NTCP) might be mainly determined by the number of ports used to deliver the valley dose. Altogether, our data suggest that MRT, currently studied on large animals (pigs [22,32] and pet animal patients), needs to be tested in a clinical environment, most likely as a multiport radiation boost delivered in “at risk” tumor regions of therapy-resistant lesions such as aggressive glioma. As a guiding principle for MRT dose prescriptions, a dual approach appears rational: separating normal tissue dose constraints tied to the valley dose prescription and antitumor effects depending on the prescribed number of ports.

## Figures and Tables

**Figure 1 cancers-13-00936-f001:**
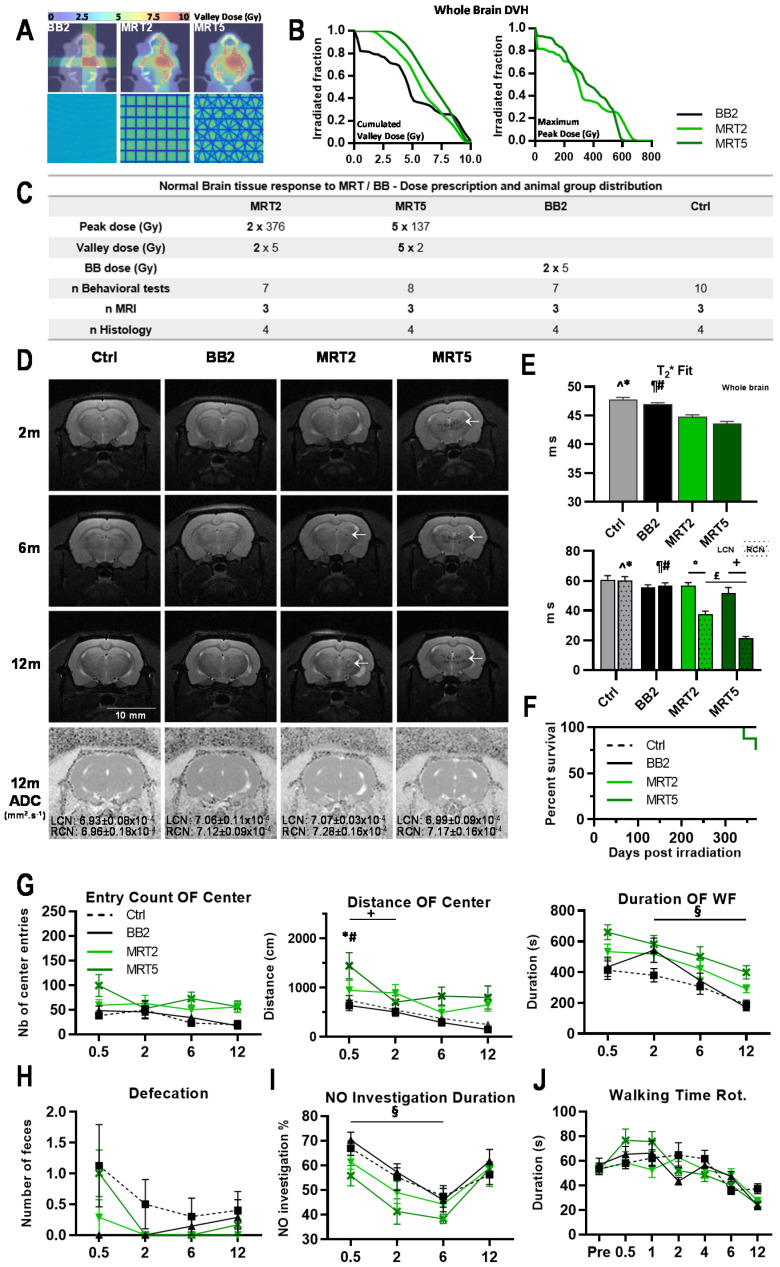
Multiple MRT irradiation focally reduced MRI signal and modified normal rat ambulation. (**A**) Irradiation geometries and valley dose maps computed on IsoGray for normal rat irradiation. BB2 was delivered as 2 orthogonal 8 × 8 mm^2^ beams (2 × 5 Gy, left panel) while MRT (19 microbeams, width 50 µm, 400 µm spacing) was delivered through 2 orthogonal (middle panel) or 5 isocentric coplanar ports spaced by 36° (right panel), intersecting in the right caudate nucleus (total cumulated valley dose of 10 Gy). (**B**) Whole brain dose–volume histograms (DVH) computed for BB2 and 2 and 5 MRT ports. Valley doses and peak doses are plotted as the cumulated dose and the maximum (cumulated intersecting microbeam doses) deposited in a 1 × 1 × 1 mm^3^ CT voxel. (**C**) Irradiation parameters and group size for animal follow-up. Ctrl: Controls. (**D**) Representative T_2_-weighted MR images acquired in normal rats at 2, 6, and 12 months after irradiation highlight hyposignal (dark horizontal markings) in the target and even in the contralateral hemisphere where multiple beams intersect. Apparent diffusion coefficients (ADC), acquired 1 year after irradiation, did not differ between the target and the contralateral caudate nucleus. (**E**) T_2_* Fit analysis unveiled reduced values in the entire brain and the target after MRT. (**F**) Kaplan–Meier curves obtained for normal rats exposed to BB2 and MRT2/5. (**G**) Open field (OF) center entry count, distance walked in center, and duration of ambulation in the whole field (WF) of normal rats at 0.5, 2, 6, and 12 months after irradiation. (**H**) Defecation of irradiated rats during the open field testing period. (**I**) Duration ratio for novel object (NO) recognition of control and irradiated rats at 0.5, 2, 6, and 12 months after irradiation. (**J**) Walking time on a turning cylinder (Rotarod, Rot.) obtained at 0.5, 1, 2, 4, 6, and 12 months after irradiation for control and irradiated rats. In each panel, control group: dashed line; BB2 group: solid black line; MRT 2 ports: light green line; MRT 5 ports: dark green line. Data are plotted as mean +/− SEM. Significance was determined using one- and two-way ANOVA tests for *p* < 0.05, and noted as * Ctrl vs. MRT5, # BB2 vs. MRT5, ^ Ctrl vs. MRT2, ¶ BB2 vs. MRT2, + MRT5 vs. MRT5, ° MRT2 vs. MRT2, £ MRT5 vs. MRT2, § BB2 vs. BB2.

**Figure 2 cancers-13-00936-f002:**
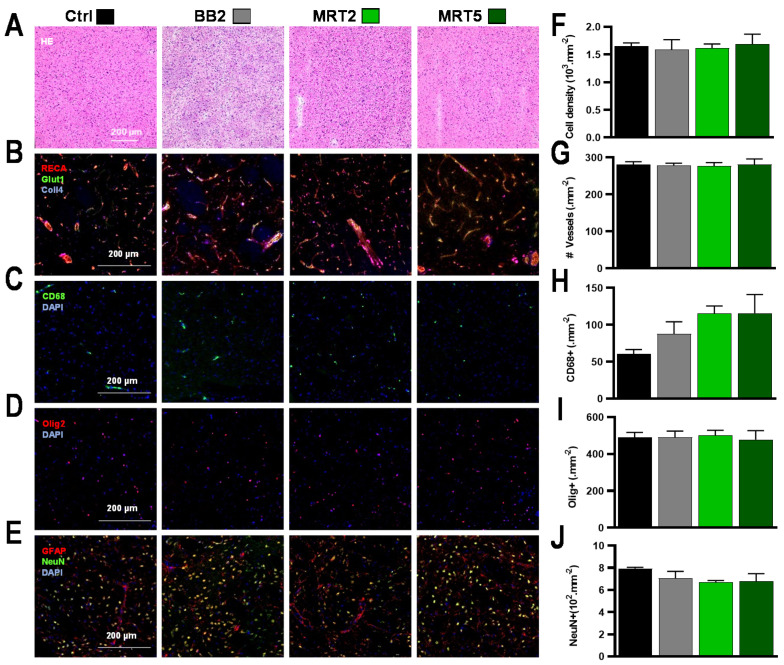
Pathology and quantitative immunolabeling characterization of irradiation effects at 12 months after irradiation of normal rats. (**A**–**I**) No histopathologic alterations were seen in collateral areas where the deposited dose was subdivided in single-beam trajectories (5 Gy BB2, 5/376 Gy MRT2 valley/peak dose, 2/137 Gy MRT5 valley/peak dose), for (**A**) H&E staining, (**B**) Collagen−4, RECA-1, Glut-1 immunolabeling, (**C**) CD68 reactivity, (**D**) Olig2 staining, and (**E**) NeuN-GFAP dual-labeling. The same results between groups were obtained for quantitative analysis of (**F**) total cell density and (**G**) number of blood vessels, while (**H**) the density of CD68-positive cells moderately increased after multiport MRT. In contrast, the same (**I**) oligodendrocyte and (**J**) neuronal densities were found in all groups.

**Figure 3 cancers-13-00936-f003:**
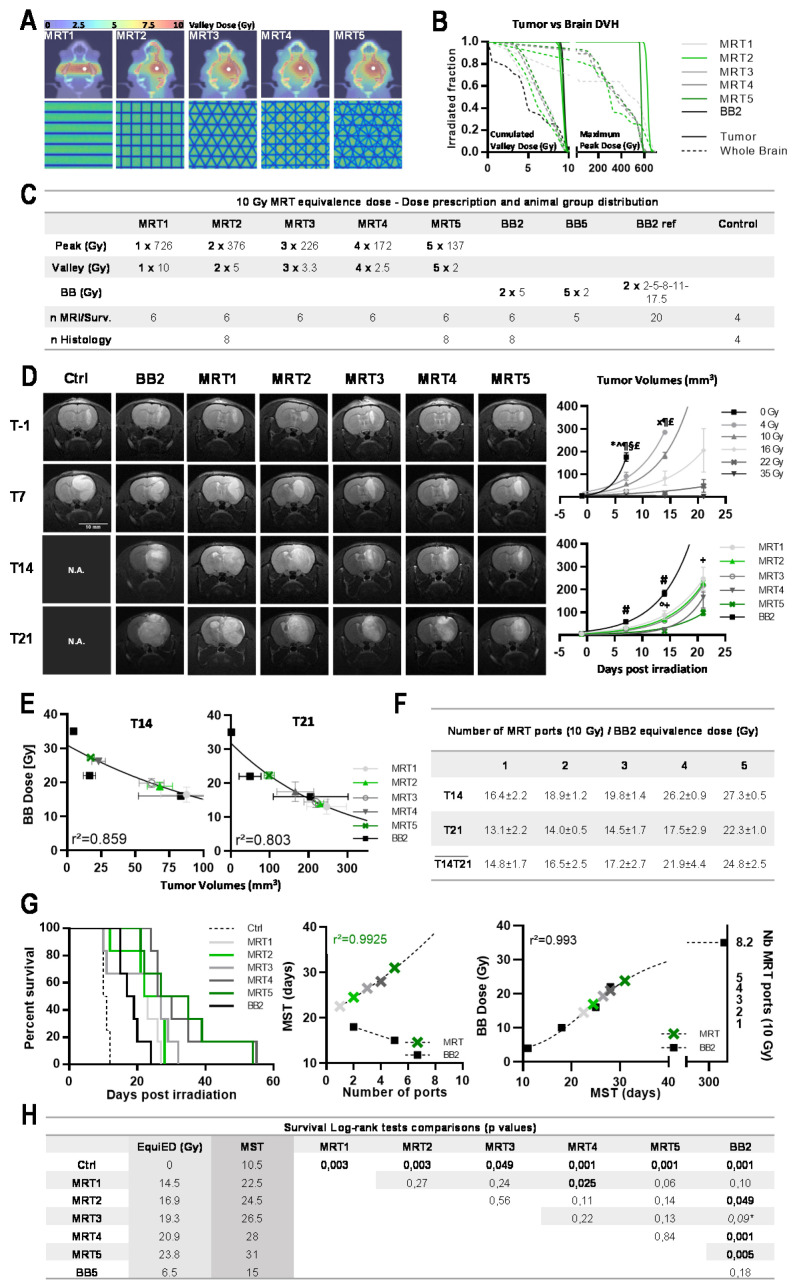
Each supplementary MRT port improved tumor control and contributed to the exponential extension of MST. (**A**) Irradiation geometries and valley dose maps computed on IsoGray for 9L glioma-bearing rats. BB2 was delivered through 2 orthogonal 8 × 8 mm^2^ beams (2 × 5 Gy) while MRT (microbeam width 50 µm, 400 µm spacing) was delivered via 1 (valley dose at target 10 Gy, peak dose 726 Gy) to 5 isocentric ports, spaced at 36° and intersecting in the right caudate nucleus (valley dose at target 5 × 2 Gy, peak dose 5 × 137 Gy). (**B**) Whole brain and tumor dose–volume histograms computed for BB2 and 1 to 5 MRT ports for a similar cumulated dose at the target (10 Gy). Valley doses and peak doses are plotted as the cumulated dose and the maximum (cumulated intersecting microbeam doses) deposited in a 1 × 1 × 1 mm^3^ CT voxel. (**C**) Irradiation parameters and group size for animal follow up. (**D**) Representative T_2_-weighted MR images acquired in 9L-bearing rats prior to and 7, 14, and 21 days after MRT irradiation. Volumes of 9L gliomas, measured on MR images at days 7, 14, and 21 after BB2 (dose range 0–35 Gy) and microbeam (10 Gy, 1 to 5 ports) irradiations, show that tumor growth control increases with use of additional MRT ports. (**E**) MRT/BB2 equivalence doses: MR tumor volumes (green) obtained at day 14 (**left**) and 21 days (**right**) after irradiation are positioned on the reference 9L tumor response curve (black line) for MRT1 to MRT5. (**F**) BB2 dose equivalences derived from (**E**) for 1 to 5 MRT ports at 14 and 21 days post irradiation and mean equivalences calculated between T14 and T21. (**G**) Survival curves of tumor-bearing rats obtained after BB2 or MRT (1 to 5 ports) for a cumulated valley dose of 10 Gy (**left**). Non-linear correlation between the number of MRT ports and MST of tumor-bearing rats (**center**). MST of 9L-bearing rats according to the delivered BB2 dose (dashed fit, **right**). By extrapolation, 8 MRT ports delivering a 10 Gy cumulated valley dose would lead to the same survival as that achieved by 35 Gy of BB2 irradiation. (**H**) Survival summary, biological equivalence doses, and Log-rank test comparisons between groups. In each panel, except D, BB2 group: solid black; MRT 1 port: light grey; MRT 2 ports: light green; MRT 3 ports: mid grey; MRT 4 ports: dark grey; MRT 5 ports: dark green. Data are plotted as mean +/− SEM. Significance was determined using unpaired *t*-tests for *p* < 0.05, and noted as ***** 0 Gy vs. all treatment groups, for BB2 groups as **^** BB 4 Gy vs. BB 16/22/35 Gy, ^x^ BB 4 Gy vs. BB 10/22/35 Gy, **¶** BB 10 Gy vs. BB 16/22/35 Gy, **§** BB 16 Gy vs. BB 35 Gy, **£** BB 22 Gy vs. BB 35 Gy and for MRT groups as **#** BB vs. all MRT groups, ° MRT4 vs. MRT1/2/3, **+** MRT5 vs. MRT1/2/3.

**Figure 4 cancers-13-00936-f004:**
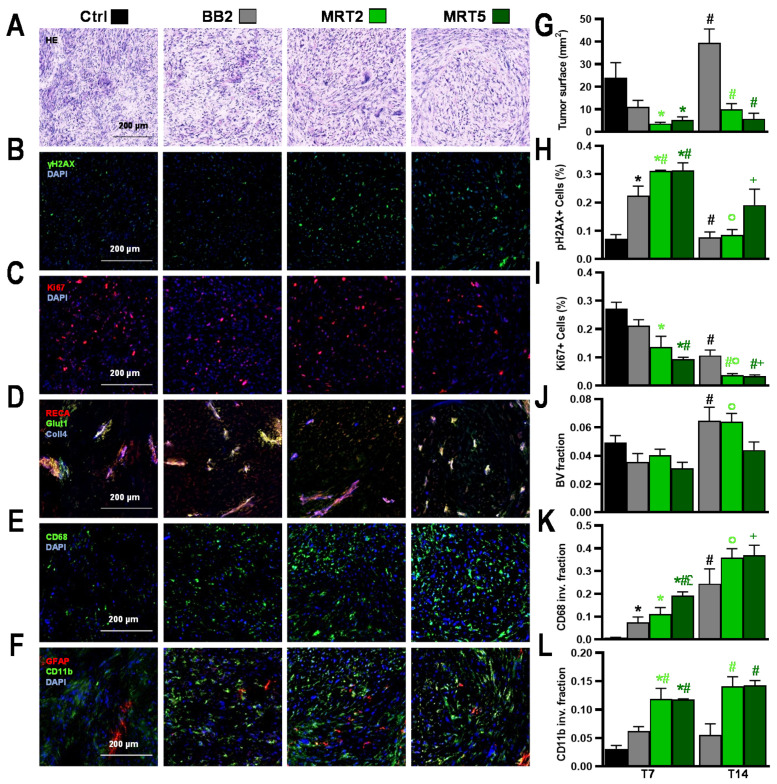
Pathology and quantitative immunolabeling characterization of irradiation effects at 7 days after 9L tumor irradiations. (**A**–**I**) Analysis of tumor lesions 7 days after BB2, MRT2, or MRT5 irradiations or 17 days after tumor implantation in untreated rats (control). (**A**) Hematoxylin and eosin staining: Irradiated tumors displayed a lower cell density than unirradiated tumors (Ctrl). While (**B**) γH2AX-reactivity was increased, (**C**) Ki67-staining decreased after MRT, in particular after MRT5. (**D**) Collagen-4, RECA-1, and GLUT-1 immuno-staining indicated vessel fractionation and hypoxia in MRT-irradiated targets. (**E**) Macrophage infiltration increased after multiport MRT as seen on CD68-stained images. (**F**) Similarly, microglia density (CD11b-positive cells) increased. Results were confirmed by quantitative analysis, showing (**G**) smaller tumors after MRT. In addition, (**H**) the γH2AX-positive cell fraction increased, whereas (**I**) the fraction of Ki67-positive cells decreased after multiport MRT. (**J**) MRT5 induced a reduction in blood volume fraction, in particular at 2 weeks p.i., compared with the other irradiation configurations. (**K**) Additionally, invasion of CD68-positive cells increased steadily, particularly after MRT5, and a delayed numerical macrophage increase was also measured two weeks after MRT2. (**L**) A similar pattern was observed for microglia invasion (CD11b-positive cell fraction). Data are plotted as mean +/− SEM. Significance was determined using multiple *t*-tests for *p* < 0.05, and noted as ***** Ctrl vs. treatment groups (black: vs. BB2; light green: vs. MRT2; dark green: vs. MRT5), **#** BB2 vs. BB2 (black) and vs. MRT (light green: vs. MRT2; dark green: vs. MRT5), **°** MRT2 vs. MRT2, **+** MRT5 vs. MRT5, **£** MRT2 vs. MRT5.

**Figure 5 cancers-13-00936-f005:**
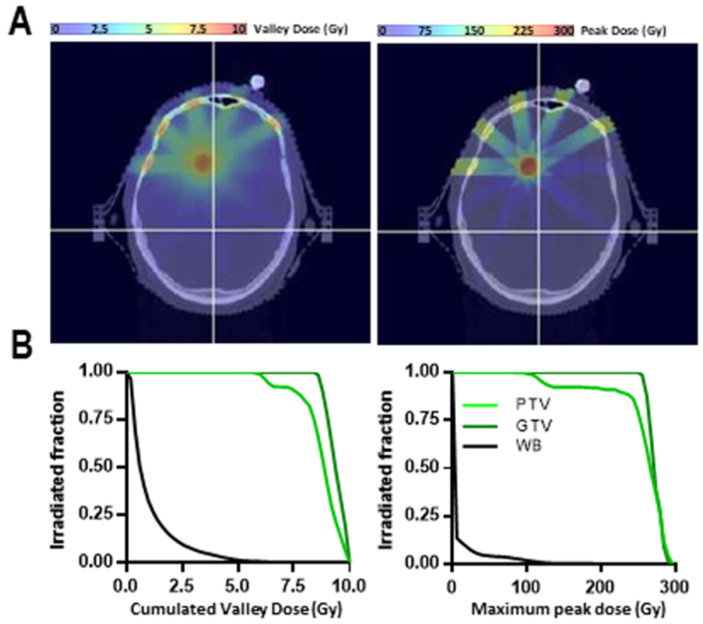
Exploratory treatment plan and provisional dosimetry for MRT (10 Gy, 5 ports) for brain metastasis irradiation for a human patient. (**A**) Valley and peak dose maps to deliver a 10 Gy cumulated valley dose to a 1 cm brain metastasis in a human patient through 5 MRT ports. (**B**) DVHs obtained for whole brain (black), PTV (light green), and GTV (dark green).

## Data Availability

The data presented in this study are available on request from the corresponding author.

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
