# Peer review of "Unexpected Benefits of Multiport Synchrotron Microbeam Radiation Therapy for Brain Tumors"

_cancers, 2021, doi:10.3390/cancers13050936_

Round 1
Reviewer 1 Report
This is a complex study that is elegantly described and brings about important and practical potential for the use of microbeams as an improved radiotherapy approach for brain and other radiation resistant tumor types. Overall it is well conceived and explained and the basic message that increased tumor response with limited increase in normal tissue effects can be obtained with educated application of MRT from multiple ports.
One area that would be appropriate to expand upon to keep the current relevance to clinical interests would be to include some thoughts/predictions about possible use in combination with immunotherapy with checkpoint antibodies. There have been some reports using spatial fractionation approaches in recent years to suggest that MRT or GRID type of therapy may preferentially stimulate anti-tumor immunity and control. This is important to touch on for perspective since the general oncology community will be wanting to know as soon as they read this compelling report.
Author Response
Answers are contained in the attached file
Best regards
Reviewer 2 Report
Please see attached document

Author Response

(The authors gave the same response as above.)

Reviewer 3 Report
This work addresses the use of multiple ports in microbeam radiation therapy. It presents a comprehensive evaluation of the effectiveness and side effects of different irradiation configurations. This is important work in investigating this novel and interesting technique and optimizing treatment strategies.
Comments:
- p3, line 99 “The number of days (n) elapsed were noted in the text as follows: Tn (p.i.).” Elapsed since when – since inoculation or since irradiation?
- p3, line 92 figure 1C of SI – I believe this should be fig S2C in the supplementary information. Also needs to be corrected on line 99.
- Section 2.4, p3, line 104-107 – “MRI of normal rats without tumors… months p.i.”. (Assuming p.i. means post-irradiation) what radiation protocols were used to irradiate these animals? Please detail here or refer the reader to the appropriate section to find this information.
- Figure 2 and section 3.1.3 – please clarify the region/irradiation circumstances of the results shown here – the figure caption states “No histopathologic alterations were seen in collateral areas”, section 3.1.3 where fig 2 is described states “The radio-induced changes in normal brain tissue located outside the intersecting regions.” – so are the sections from normal brain tissue outside the target but still within a single irradiation field?
- Line 246, p6 “A BB irradiation of 10Gy with 2 cross-fired beams significantly improved MST, from 10.5 to 18 days p.i.” – improved compared to what?
- Section 3.3 – what field size has been used in planning the patient, for comparison with the setup used for the animal work (“8x8 mm2 irradiation field, 19 96 microbeams, width 50μm, spacing 400μm”, Line 96)?
- The broad beam geometry used appears to always be 2 opposing beams, but is sometimes referred to as BB and sometimes BB2. Please be consistent; I would advise using BB2 unless deliberately referring more widely to all broad beam treatments.
Minor/typos:
- Line 100 “…treatment plan has been performed on a patient…” sounds as if a patient has been treated with the plan. Perhaps better to word as “… treatment plan has been calculated for a patient…”.
- Line 103 “…histograms dose volume..” should be “…dose volume histograms..” Similarly, line 237 p6 – “HDV” should be “DVH”.
- Fig 3G, centre – “Number of incidences” axis title is a little confusing, perhaps better as “Number of ports”?
- It would be nice to label Fig1A, Fig3A with “MRT1”, “MRT2” etc.
- Line 312-3 “microglial cells did not significantly infiltrate 312 9L tumors.” should include a reference to Fig 4F/L.
Author Response

(The authors gave the same response as above.)
